# Differences in Biofilm Formation by Methicillin-Resistant and Methicillin-Susceptible *Staphylococcus aureus* Strains

**DOI:** 10.3390/diseases11040160

**Published:** 2023-11-05

**Authors:** Eduardo Hernández-Cuellar, Kohsuke Tsuchiya, Ricardo Valle-Ríos, Oscar Medina-Contreras

**Affiliations:** 1Laboratorio de Biología Celular y Tisular, Departamento de Morfología, Universidad Autónoma de Aguascalientes, Aguascalientes 20100, C.P., México; 2Division of Immunology and Molecular Biology, Cancer Research Institute, Kanazawa University, Kakuma-machi, Kanazawa 920-1192, Japan; ktsuchiya@staff.kanazawa-u.ac.jp; 3Research Division, Faculty of Medicine, Universidad Nacional Autónoma de México (UNAM), Mexico City 04360, C.P., México; vallerios@unam.mx; 4Laboratory of Research in Immunology and Proteomics, Federico Gómez Children’s Hospital of Mexico, Mexico City 06720, C.P., México; 5Epidemiology, Endocrinology & Nutrition Research Unit, Mexico Children’s Hospital (HIMFG), Mexico City 06720, C.P., México; omedina@himfg.edu.mx

**Keywords:** MRSA, MSSA, biofilms, PIA (polysaccharide intercellular adhesin), *icaADBC* operon, Agr, SarA

## Abstract

*Staphylococcus aureus* (*S. aureus*) is a common pathogen involved in community- and hospital-acquired infections. Its biofilm formation ability predisposes it to device-related infections. Methicillin-resistant *S. aureus* (MRSA) strains are associated with more serious infections and higher mortality rates and are more complex in terms of antibiotic resistance. It is still controversial whether MRSA are indeed more virulent than methicillin-susceptible *S. aureus* (MSSA) strains. A difference in biofilm formation by both types of bacteria has been suggested, but how only the presence of the SCC*mec* cassette or *mecA* influences this phenotype remains unclear. In this review, we have searched for literature studying the difference in biofilm formation by MRSA and MSSA. We highlighted the relevance of the *icaADBC* operon in the PIA-dependent biofilms generated by MSSA under osmotic stress conditions, and the role of extracellular DNA and surface proteins in the PIA-independent biofilms generated by MRSA. We described the prominent role of surface proteins with the LPXTG motif and hydrolases for the release of extracellular DNA in the MRSA biofilm formation. Finally, we explained the main regulatory systems in *S. aureus* involved in virulence and biofilm formation, such as the SarA and Agr systems. As most of the studies were in vitro using inert surfaces, it will be necessary in the future to focus on biofilm formation on extracellular matrix components and its relevance in the pathogenesis of infection by both types of strains using in vivo animal models.

## 1. Introduction

*Staphylococcus aureus* is a common human pathogen capable of producing community- and hospital-acquired infections. The severity of infections varies from mild skin and soft tissue infections to more complicated ones, such as bacteremia, osteomyelitis, and endocarditis [1]. Methicillin, a β-lactam antibiotic that targets the biosynthesis of bacterial cell walls, was used as an alternative to replace penicillin, as it was known in the 1950s that most *S. aureus* isolates were resistant to this antibiotic. However, the first cases of methicillin resistance were detected as early as 1 year after its introduction in 1959 [2]. Even though methicillin is no longer a commercially available antibiotic, the concept of methicillin-resistant *S. aureus* (MRSA) is still in use and refers to virtually all the *S. aureus* strains resistant to β-lactam antibiotics, except for the latest generation of cephalosporines [1]. MRSA strains have been a serious worldwide human threat as their infections are more difficult to treat in comparison with those of methicillin-susceptible *S. aureus* (MSSA). Methicillin resistance in *S. aureus* is associated with the presence of the *mecA* gene that encodes the protein PBP2a (penicillin-binding protein 2a) with lower affinity to β-lactam antibiotics. This type of resistance is not acquired due to the presence of a plasmid containing an antibiotic resistance gene but due to the incorporation of a foreign mobile genetic element containing *mecA* called a staphylococcal cassette chromosome (SCC*mec*) into the bacterial chromosome of MRSA strains [3].

Since the discovery of the *mecA* gene as responsible for the resistance to most β-lactam antibiotics in MRSA isolates, and its association with higher mortality rates [2,4], interest in the virulence factors, resistance to other antibiotics, and the capacity of biofilm formation by these strains has emerged. Even though there is not a direct association of the *mecA* gene or *SCCmec* element with other genes conferring resistance to other antibiotics or biofilm-related genes, it may be possible that MRSA isolates possess a more complex genetic background in comparison with MSSA. However, there are still controversial results in terms of the difference in the virulence between both types of bacteria, and it may be due to the diversity of clones, making this analysis not that simple [5,6].

We have searched in the databases PubMed, Scopus, and Web of Science for publications analyzing the differences in biofilm formation by MRSA and MSSA isolates using similar conditions. We found that from 20 research articles [7,8,9,10,11,12,13,14,15,16,17,18,19,20,21,22,23,24,25,26], 10 (50%) mentioned that MRSA isolates were better at biofilm formation than MSSA, 9 (45%) found no differences, and only 1 (5%) reported that MSSA were better than MRSA (Table 1). For biofilm formation analysis, the microtiter method was used with crystal violet (CV) staining or a similar method (65%), or the Congo red method was used together with the microtiter method (35%) (Appendix A). The Congo red method is a qualitative assay for the detection of biofilm-producing bacteria. The method uses Congo red dye and sucrose in BHI agar. Bacteria forming black colonies with a dry and crystalline appearance are considered biofilm producers, while those with pinkish-red colonies are consider non-biofilm producers. In case of the microtiter plate assay, the bacteria are allowed to form biofilms in 96-well polystyrene microplates, and then the biofilm is stained with crystal violet dye. Finally, the biofilm can be quantified by dissolving it with acetic acid and checking the optical density at around 600 nm in a plate reader [27]. It is worth mentioning that there are many factors that may have influenced the results: for example, for the microtiter method, Tryptic Soy Broth was the predominant medium used; however, supplementation with glucose varied in concentration, from no supplementation to 0.25, 0.5, 1, or 2% %) (Appendix A). Also, there was variation related to the way of classifying the ability to form biofilms; for example, in several publications, it was reported as a classification from non-producers to weak, moderate, or strong biofilm producers. In other cases, it was only mentioned if the isolates were able to produce biofilm or not, or a total quantification of the biofilm biomass was undertaken by measuring the optical density. The percentage of correspondence between the Congo red method and the microtiter method with CV staining was only mentioned in three publications, with an agreement between both methods of 96%, 98%, and 28.2% [10,17,26]. As expected, in the work with the lowest percentage of agreement, it was found that MRSA isolates were better biofilm producers than MSSA isolates only for the Congo red assay but not for the microtiter method [10]. Furthermore, another form of variation in the results may have come from the statistics analysis used; for example, it was shown that for MSSA, 21 out of 36 (58.33%) isolates were able to form biofilm, while for MRSA, it was 46 out of 56 (82.14%). However, the differences were reported as not statistically significant [23]. Finally, there were also differences in the way of classifying the *S. aureus* isolates as MRSA or MSSA in the 20 publications. In 8 (40%), disk diffusion or a similar method to analyze the susceptibility to cefoxitin or oxacillin in accordance with the Clinical and Laboratory Standards Institute (CLSI) guidelines was mentioned, and in 12 (60%), the disk diffusion method was used together with the detection of the *mecA* gene (Table 1). For the antibiotic susceptibility analysis, nine publications mentioned the detection of cefoxitin (45%), four oxacillin (20%), two both antibiotics (10%), and five did not specify which antibiotic was used (25%) (Table 1). It is worth noting that in the nine publications that reported no differences between the MRSA and MSSA isolates in terms of biofilm formation, six used only the disk diffusion method to classify the *S. aureus* isolates as MRSA, while in the cases where MRSA was better at biofilm formation, 8 out of 10 used *mecA* detection and antibiotic susceptibility analysis. We suggest that confirmation with *mecA* detection is important in the classification of *S. aureus* as MRSA. In this regard, it was reported that from 15 MRSA isolates, only 8 (53%) were positive for the presence of *mecA*, while all the MSSA isolates analyzed were negative [7]. It is generally accepted that methicillin resistance is acquired due to the presence of *mecA*; however, there are descriptions of methicillin resistance independent of *mecA*, such as borderline oxacillin-resistant *S. aureus* (BORSA) strains and modified methicillin-resistant *S. aureus* (MODSA) strains, but their frequency and clinical relevance is uncertain [28,29].

Based on this analysis, there is not enough evidence to support any difference in biofilm formation between the MRSA and MSSA isolates. Also, the results may be validated with more precise lineage classification of the isolates, as a dominant clone may have been present during the time of the study in a specific region. For example, there have been described for *S. aureus* several clonal complexes and endemic or global distributions based on the differences in the sequence of seven housekeeping genes (*arcC*, *aroE*, *glpF*, *gmk*, *pta*, *tpi*, and *yqiL*), as discovered using multi-locus sequence typing (MLST) (S. aureus clonal complex designation|PubMLST) [30]. These clonal complexes, depending on the type, may include community-associated MRSA (CA-MRSA), hospital-associated MRSA (HA-MRSA), or MSSA clones. In addition, 14 types of SCC*mec* elements are currently recognized by the International Working Group on the Classification of Staphylococcal Cassette Chromosomal Elements based on the genetic variation of the two main components, the ccr gene complex and the mec gene [31]. Some SCC*mec* types have been associated with HA-MRSA (SCC*mec* type I, II, and III), while others with CA-MRSA (SCC*mec* type IV and V) [32]. Therefore, the association analysis between MRSA and MSSA isolates in relation to biofilm formation capacity, must include in the future a better characterization of the isolates in order to provide more information related to their clonal origin.

### 1.1. Differences in Biofilm Formation by MRSA and MSSA

PIA (Polysaccharide intercellular adhesin), also called PNAG (poly-N-acetylglucosamine), is a glycan with repeats of β-1,6-linked 2-acetamido-2-deoxy-D-glucopyranosyl residues, involved in intercellular adhesion, aggregation, and attachment to abiotic surfaces for *S. aureus*. PIA synthesis depends on the *ica*(intercellular adhesin)-*ADBC* operon, whose expression products are IcaA, IcaD, IcaB, and IcaC [33]. IcaA is a transmembrane *N*-acetyl-glucosaminyltransferase that, together with IcaD, synthesizes *N*-acetyl-glucosamine oligomers of more than 20 residues. Then, IcaC translocates these polysaccharides out of the cell surface. Finally, IcaB deacetylates the poly-*N*-acetylglucosamine molecules, which is important for the adherence of these polymers to the outer cell surface and biofilm formation (Figure 1) [33]. The role of PIA and the *ica* operon in biofilm formation was first described in *Staphylococcus epidermidis* (*S. epidermidis*) [34]. Later, Crampton et al. showed for the first time the presence of the *ica* operon in all the strains of *S. aureus* analyzed. Also, the mutation of the *ica* locus in the *S. aureus* strain ATCC 35556 resulted in reduced biofilm formation [35]. Since then, PIA has been considered the main extracellular matrix component involved in biofilm formation by *S. aureus*. However, it is worth noting that not all the strains analyzed by Crampton were able to form biofilm in vitro [35]. Likewise, it was tested the ability to form biofilm by 128 *S. aureus* isolates, and only 57.1% showed a biofilm-positive phenotype, even though all of them were *icaA*-positive [36]. We now know that *icaADBC* genes are present in most of the *S. aureus* clinical isolates, and that the expression of those genes is tightly regulated in vitro; for example, CO_2_ levels, anaerobicity, glucose, and osmotic stress have been shown to influence *ica* operon expression and biofilm formation [19,37]. This may explain, at least in part, why not all the *S. aureus* isolates containing those genes can form biofilms. 

Fitzpatrick et al. showed differences in biofilm formation by MRSA and MSSA clinical isolates using BHI medium alone or supplemented with 1% glucose or 4% NaCl. A higher percentage of MSSA isolates were able to form a biofilm in a NaCl medium in comparison with MRSA, while the proportion was similar in BHI glucose. It was suggested a possible association between methicillin susceptibility and biofilm formation depending on the specific environmental conditions [37]. It is worth mentioning that the association of biofilm production, methicillin susceptibility, and *mecA* expression was first analyzed in *S. epidermidis* [38,39]. Also, the transcription of the *ica* operon and biofilm formation induced by NaCl was first reported in *S. epidermidis* [40,41]. It is now well accepted that the *ica* operon is expressed under osmotic stress conditions, and this correlates with biofilm formation in MSSA strains [42]. In case of MRSA, it was shown an increase in biofilm formation by four *ica*-positive MRSA strains in a BHI medium supplemented with glucose but not in a BHI or BHI-NaCl medium [37]. Also, only in one MRSA strain was the expression of *icaA* induced in BHI-NaCl, but the deletion of the *ica* operon in this strain did not affect the biofilm formation in BHI-glucose. The expression of *ica* operon at the transcriptional level was not dependent on the presence of NaCl for the others three MRSA strains. The authors concluded that the ability to form biofilm only in BHI-glucose by these MRSA strains was independent of the *ica-ADBC* operon [37]. 

Later, the ability of MRSA and MSSA strains isolated from device-related infections to form biofilms was tested. Again, it was found that biofilm formation in BHI-NaCl was more frequent in MSSA strains, in comparison with MRSA, which preferentially formed biofilms in a medium containing glucose. For most of the MSSA strains tested, as expected, the addition of NaCl to the BHI medium was associated with the expression of *icaA*, PIA production, and biofilm formation. Even though the *icaA* expression was also increased in most MRSA strains in the presence of NaCl, PIA production was not observed [19], suggesting post-transcriptional regulation of PIA synthesis. Furthermore, deletion of the *ica* operon in the MSSA isolates affected their ability to form biofilms in BHI NaCl, while deletion of the *ica* operon in the MRSA isolates did not affect their ability to form biofilms in BHI glucose. Interestingly, glucose-induced MRSA biofilms were susceptible to treatment with proteinase K, while NaCl-induced MSSA biofilms were susceptible to sodium periodate, a reagent used to destabilize β-1,6-linked polysaccharides [19]. Thus, the chemical composition of the biofilms generated by both types of strains under those conditions seems to be different. 

### 1.2. Biofilm Formation Independent of the icaADBC Operon

In addition to the aforementioned cases of biofilm formation independent of the *ica* operon in MRSA strains in a medium containing glucose, one of the first reports of an *ica*-independent biofilm formation in *S. aureus* was from a clinical isolate named UAMS-1 (MSSA). It was shown that a mutation in the *ica* locus had little impact on its biofilm formation capacity in vitro and in vivo [43]. Also, Lim et al. identified using mutagenesis analysis a gene called *rbf* involved in biofilm formation. The sequence suggested that it was a transcriptional regulator, and it did not affect the expression of the *ica* operon. Thus, it was considered an *ica*-independent mechanism [44]. Upstream of the *ica* operon is the *icaR* gene; Conlon et al. showed that icaR is a repressor of PIA production and biofilm formation by inhibiting the expression of the *ica* operon in *S. epidermidis* [40]. It was later shown that Rbf promotes biofilm formation by repressing the transcription of *icaR* in *S. aureus* [45]. 

Also found using mutagenesis analysis was a gene called *bap* (biofilm-associated protein) involved in attachment to inert surfaces, intercellular adhesion, and biofilm formation. Bap was described as a surface protein consisting of 2276 amino acids. However, it was only found in a small fraction of the *S. aureus* isolates from bovine mastitis lesions (5%), while it was absent in the clinical *S. aureus* isolates analyzed from humans [46]. In the *S. aureus* strain SA113, whose biofilm formation ability has been associated with the *ica* operon, complementation with the *bap* gene in the mutant strain SA113 Δ*ica* resulted in biofilm formation, suggesting the role of bap in biofilm formation independent of the *ica* operon [46]. In fact, bap is a surface protein containing the LPXTG motif. Since biofilms from MRSA strains in BHI-glucose were susceptible to proteinase K treatment, it is reasonable to believe that surface proteins such as bap must be responsible for the *ica*-independent biofilm formation in these strains. In this regard, a genomic analysis from MRSA strains revealed the presence of several LPXTG surface proteins presumed to be involved in the adherence to host tissues such as clumping factor A and B (Clf A/B), fibronectin binding protein A and B (Fnb A/B), collagen adhesin (Cna), protein A (Spa), methicillin resistance surface protein (Pls), SdrC/D/E, and several cell wall surface anchor proteins (Sas A/B/C/D/E/F/G/H/I/J/K). There were also found other non-LPXTG surface proteins such as autolysin (Atl), elastin binding protein (Ebp), and the fibrinogen binding proteins Fib and Efb [47].

Likewise, a mutation of sortase A (SrtA), an enzyme that catalyzes the cell wall anchoring of LPXTG proteins, in an *S. aureus* strain with an *ica*-independent biofilm phenotype was deficient in biofilm formation [48,49]. It was also shown using mass spectrometry that SdrD and protein A (Spa) were key components of the cell wall of an *S. aureus* strain able to form *ica*-independent biofilms, and deletion of the *spa* gene resulted in reduced biofilm formation [48]. SasC and SasG were also shown to promote biofilm formation in *S. aureus* [50,51]. Interestingly, it was found that MRSA strains in a BHI medium with glucose lowered the pH to mildly acidic conditions, and this was associated with the expression of *fnbA* and *fnbB*, and biofilm formation [49]. Mutations of *fnbA* and *fnbB* in several MRSA strains reduced biofilm formation. However, these mutations did not affect the biofilm formation in MSSA strains with a PIA-dependent biofilm phenotype. It was proposed that the effect on biofilm formation of *fnbA* and *fnbB* was due to intercellular aggregation rather than initial attachment [49]. Furthermore, carriage of both the *fnbA* and *fnbB* genes correlated with higher biofilm formation in MRSA strains in comparison with those strains carrying only one gene. It is worth mentioning that the proportion of bacteria expressing both genes was even higher in the MSSA clinical isolates, 69% (68/99) for MSSA, and 33% (40/118) for MRSA [52]. Thus, LPXTG proteins seem to be important in the proteinaceous biofilms formed by MRSA strains.

Also described was extracellular DNA (eDNA) as an important component of the *S. aureus* biofilms, as addition of DNase I to the medium or to the preformed biofilms resulted in their inhibition or destabilization, respectively [53,54]. CidA, a hydrolase involved in cell lysis, was shown to be important for DNA release and biofilm regulation in *S. aureus* [54]. Later, the major autolysin of *S. aureus* called Atl was found to mediate cell lysis and eDNA release, contributing to biofilm formation [55]. It was also shown that Atl was important for attachment and biofilm formation in MRSA strains with a FnbA/B biofilm phenotype, but not for MSSA strains with a PIA-dependent biofilm phenotype [56]. Furthermore, mutation of the protease ClpP in the *S. aureus* Newman strain resulted in an increase in biofilm formation, and it was a PIA-independent but protein- and eDNA-dependent biofilm [57]. Transcription of the *agrA* and *agrC* genes was reduced in the *clpP*-mutant strain, and it was associated with biofilm formation and decreased levels of protease activity. Interestingly, an increase in the expression of the hydrolase *sle1* was found in the *clpP*-mutant strain, together with an increase in cell lysis and eDNA release [57]; however, it was not clear how the Agr system was affected in the *clpP*-mutant strain. Altogether, these results indicate the relevance of hydrolases in eDNA release and their contribution to biofilm formation by *S. aureus*. It is now well accepted that the biofilm composition in MRSA strains depends more on surface proteins and eDNA in comparison with the *ica*-dependent biofilm phenotype in MSSA strains [42]. Most of the studies have used a medium with NaCl for MSSA strains and a medium with glucose for MRSA strains, favoring the mentioned biofilm phenotypes. However, it is worth noting that a high proportion of MSSA isolates can also form biofilms in mediums containing glucose [19,20]. Thus, it is reasonable that depending on the environment, MSSA strains could form biofilms with a PIA-independent phenotype as well. Indeed, the first experiments analyzing the content of eDNA in the biofilms of *S. aureus* used UAMS-1 and SH1000, two MSSA strains [53,54,55].

Interestingly, a MRSA clinical isolate capable of switching from a PIA-dependent to a PIA-independent biofilm was found, depending on the medium composition. For example, in a TSB medium with glucose, it formed a PIA-independent biofilm, but when NaCl was added to the medium, detection of PIA and biofilm formation was observed. Deletion of *ica* in this *S. aureus* strain did not affect its capacity to form a biofilm in a medium with glucose, but it completely lost the capacity of biofilm formation in a medium with NaCl [58]. Sortase A (*srtA*) mutation revealed that LPXTG proteins were important for biofilm formation in a medium with glucose while they did not affect the generation of a PIA-dependent biofilm in a medium with NaCl. Genomic analysis revealed a unique repertoire of 20 LPXTG proteins in this strain, and deletion of *fnbA* and *fnbB* was associated with a complete loss of the PIA-independent biofilm. Also, *fnbB* was induced at the transcriptional level in the medium with glucose in comparison with the medium containing NaCl, but the *icaC* mRNA levels were unaffected in both types of media, suggesting again that there must exist post-transcriptional regulatory mechanisms in PIA expression [58]. 

### 1.3. Main Regulatory Systems in Biofilm Formation by S. aureus

The accessory gene regulator (agr) operon consists of the *agrBDCA* genes and is part of a quorum-sensing regulatory system in *S. aureus*. The expression of many virulence factors in *S. aureus* is controlled by the Agr system, which consists of two transcripts whose expression are controlled by promoters P2 and P3. The first transcript is an operon that encodes the four *agrBDCA* genes, and the second is RNAIII, the true effector molecule that regulates the expression of all the genes regulated by the Agr system. The gene encoding the δ-toxin (*hld*) is in the effector RNAIII regulatory molecule. This system is activated during the transition from the exponential growth phase to the stationary growth phase [59]. The Agr system responds to the extracellular levels of AIP (autoinducing peptide), an eight-residue peptide, with the last five residues forming a thiolactone cyclic ring. AIPs are formed via the action of AgrB, a membrane-bound peptidase that proteolytically processes AgrD, the peptide precursor of AIP. AgrC is a membrane-bound histidine kinase sensor of AIPs that is autophosphorylated, and this signal passes to the response regulator AgrA, which in turn binds to the promoters P2 and P3 to start their transcription (Figure 2) [60]. 

Analyzing the agr phenotype in terms of δ-toxin production (the product of the RNAIII transcript in the *agr* operon) and the biofilm formation capacity, it was demonstrated that 78% of the *agr*-negative *S. aureus* strains formed a biofilm in comparison with only 6% of the *agr*-positive strains [59]. It was also shown that *agr* mutation in *S. aureus* strains enhanced biofilm formation in comparison with the wild-type strains. Interestingly, the expression of PIA was not regulated by the Agr system [59]. It has been suggested that *agr* is repressed in a medium containing glucose and it correlates with biofilm formation [42]. Indeed, it was shown that expression of RNAIII was repressed in a medium containing glucose [61]. It was also found that reactivation of the Agr system via AIP addition or glucose depletion triggered detachment, destabilizing biofilms, only in *agr*-competent strains [61]. It has been shown that the activation of the Agr system results in upregulation of the Aur metalloprotease and the SpIABCDEF serine proteases, together with a downregulation of several surface proteins [62]. In fact, the inhibition of serine proteases and an *S. aureus aur* mutant, or a double mutant strain deficient in the *aur* gene and *SpI* genes, showed reduced detachment and a stable biofilm after the addition of AIPs, indicating that the Agr system destabilizes biofilms via the action of extracellular proteases. 

Furthermore, Beenkem et al. described an *S. aureus* strain RN6390 that was unable to form biofilm in vitro, and deletion of the *agr* locus resulted in gaining the ability to form biofilm for this strain. However, a second deletion of *sarA* in the RN6390 *agr*-deficient strain resulted again in a lack of biofilm formation, suggesting an association between these genes [63]. Indeed, SarA (staphylococcal accessory regulator) is a DNA-binding protein regulating the transcription of genes important for virulence in *S. aureus* such as *agr*, *hla*, *spa*, and *fbnA*, indicating *agr*-dependent and *agr*-independent regulatory mechanisms in SarA [64]. For example, it was described using a *sarA*- or *agr*-mutant strain that the transcription of 104 genes was upregulated and 34 downregulated in an *agr*-dependent manner, while 76 were upregulated and 44 downregulated in a *sarA*-dependent manner [62]. It was concluded that *sarA* transcriptional regulation is not limited only to the *agr* genes. It was also described that SarA represses the collagen adhesin gene (*cna*) in an *agr*-independent manner [65]. Similarly, it was found that SarA positively controlled bap-dependent biofilm formation in an *agr*-independent mechanism in three *S. aureus* strains, and it was shown that SarA binds to the bap promoter, regulating its expression [66]. Also, proposed was the *agr*-independent transcriptional regulation of *fbnA* by the binding of SarA to the promoter site of this gene [67]. Thus, there is enough evidence of the SarA regulation of genes related to biofilm formation in *S. aureus* independent of the Agr system.

In addition, it was shown that *sarA* mutants decreased PIA production and biofilm formation by downregulating *ica* operon transcription. Furthermore, *sarA* mutants showed higher protease activity beyond the downregulation of PIA synthesis [68,69,70]. It was also found in *S. aureus* clinical isolates amenable to genetic manipulation that biofilm formation was abolished for *sarA* MRSA mutants in a glucose-containing medium and for *sarA* MSSA mutants in a NaCl-containing medium [19]. This indicates the role of SarA in PIA-dependent and PIA-independent biofilm formation. In comparison with *agr* mutants, only 5 out of 21 of the *agr* MRSA mutants resulted in an increase in biofilm formation, highlighting the role of the Agr system in some of the PIA-independent proteinaceous biofilms of MRSA strains, rather than in the PIA-dependent biofilms of MSSA strains [19]. 

It was also described for *S. epidermidis* that mutation in *σ^B^* (a stress response regulator, SigB) or *rsbU*, an activator of *σ^B^*, results in reduced *ica* transcription and biofilm formation in osmotic stress conditions, which favored the PIA-dependent biofilm phenotype. It was found that *σ^B^* acts as repressor of the negative regulator *icaR* gene [71]. Interestingly, it was shown in *S. aureus* that SarA and *σ^B^* induced the expression of *icaR* beyond that of the *icaADBC* operon [72]. As expected, an *S. aureus* double mutant lacking the expression of *sarA* and *σ^B^* showed an enhanced reduction in the transcriptional expression of the *ica* operon. Surprisingly, the biofilm formation was only partially affected in the double mutant in comparison with the *sarA* mutant. Also, the *σ^B^* mutant was not affected in its biofilm formation and PIA production [68]. This unexpected result unveils a complex regulation of biofilm-related genes by SarA and SigB that may be independent of each other.

Finally, LuxS is an enzyme that catalyzes the production of a signaling molecule called autoinducer-2 (AI-2). The LuxS/AI-2 quorum-sensing system is important for bacterial growth, metabolism, virulence, and biofilm formation. It was shown that the LuxS/AI-2 system negatively regulates PIA production and biofilm formation in *S. aureus* by repressing the transcription of *rbf* [73]. We have previously mentioned that Rbf represses the transcription of *icaR*, the negative regulator of the *ica* operon [45].

### 1.4. Concluding Remarks and Future Directions

We have addressed in this review the recent literature comparing the ability of biofilm formation by MRSA and MSSA isolates. MRSA isolates are resistant to most of β-lactam antibiotics, but whether MRSA strains possess different virulent factors that may further complicate infections is still controversial. In fact, it seems to be that the presence of virulence factors is associated with the origin of the isolates. In this regard, we explained that the bap protein was found in *S. aureus* isolates from bovine mastitis lesions while it was absent in isolates from human origin. Furthermore, tracking the origin of clones for epidemiology studies has been accomplished by analyzing the clonal complexes and the type of SCC*mec* elements in MRSA strains. Even though in some studies, the association of biofilm formation with a specific type of SCC*mec* element has been proposed, we consider that the origin of the isolates, rather than the mere acquisition of the SCC*mec* element, is important for this phenotype. Interestingly, the finding of MSSA isolates forming biofilms preferentially via a PIA-dependent mechanism, and MRSA isolates forming them via a PIA-independent but protein- and eDNA-dependent mechanism, suggests differences in biofilm formation by these strains. We highlighted the role of several hydrolases involved in the release of eDNA and their contributions to biofilm formation by MRSA strains, as well as surface proteins containing the LPXTG motif, which are anchored to the cell wall via sortase A. Genetic analysis revealed the presence of a unique repertoire of these types of proteins in MRSA strains, and functional analysis showed that some of these proteins are in fact important for biofilm formation depending on the MRSA strain analyzed. Finally, the gene regulatory protein SarA is not only an important regulator of PIA-dependent biofilms by controlling the expression of the *ica* operon, but also induces the expression of the Agr system, which in turn regulates PIA-independent biofilm formation by modulating the expression of extracellular proteases and surface proteins. As most of the research analyzing the biofilm formation capacity in *S. aureus* has been undertaken in vitro on abiotic surfaces, we consider that in the future, the contribution of the relevant genes that have been described in the context of adhesion and biofilm formation on extracellular matrix components should be analyzed, as well as their relevance to pathogenesis, using in vivo infection models.

## Figures and Tables

**Figure 1 diseases-11-00160-f001:**
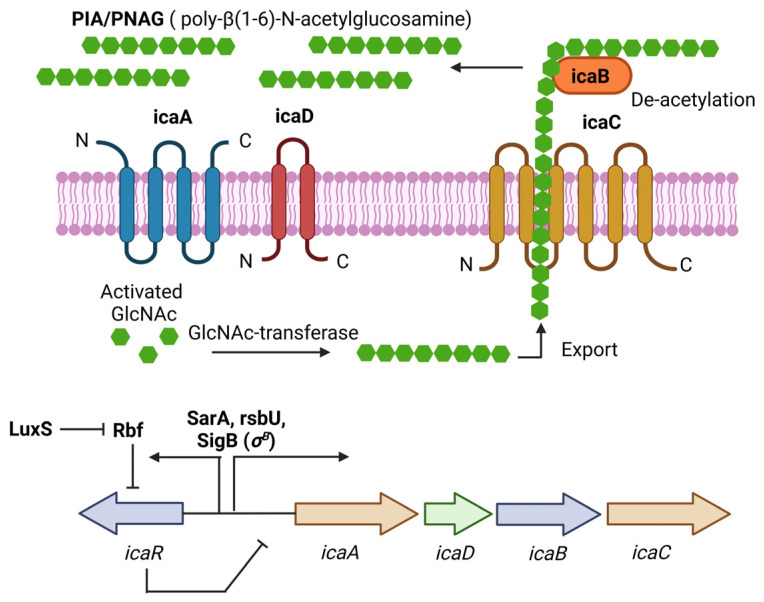
PIA (polysaccharide intercellular adhesin) is produced by the *icaADBC* operon in *S. aureus*. PIA-dependent biofilm formation has been described mainly for MSSA isolates under osmotic stress conditions. The *ica* operon is regulated by SarA and SigB. IcaR is part of the operon and acts as a negative regulator of PIA production. Figure was created using BioRender.com.

**Figure 2 diseases-11-00160-f002:**
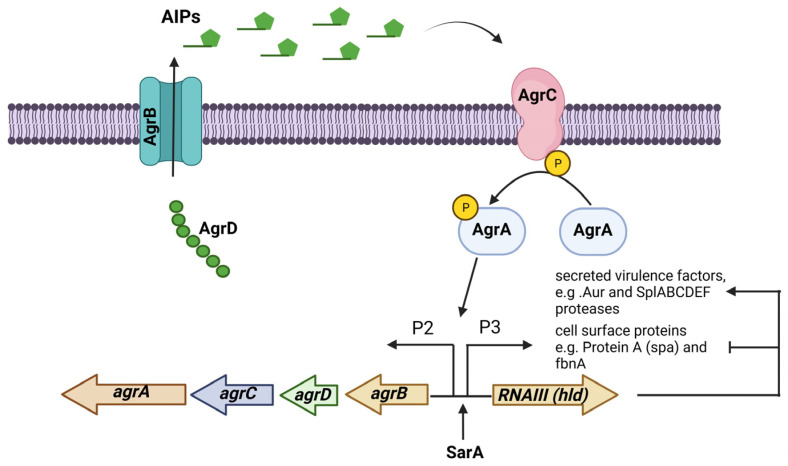
The Agr (accessory gene regulator) system is an operon that regulates the expression of several virulence factors in *S. aureus*. The *agrBDCA* operon is activated by SarA. In some *S. aureus* strains, the Agr system regulates PIA-independent biofilm formation by negatively regulating the expression of surface proteins while positively regulating the expression of extracellular proteases. Figure was created using BioRender.com.

**Table 1 diseases-11-00160-t001:** Percentage of publications reporting that MRSA isolates had better (>), less (<), or the same (=) biofilm formation capacity in comparison with MSSA. Characterization of MRSA isolates was assessed via susceptibility to cefoxitin and/or oxacillin using the disk diffusion method (DDM) or *mecA* detection together with the DDM. For the DDM, the antibiotic is indicated as cefoxitin (Cef), oxacillin (Ox), not mentioned (Nm), and not done (Nd).

	MRSA > MSSA	MRSA = MSSA	MRSA < MSSA
*mecA* + DDM	40% (8/20) REF. [7,8,11,14] (Cef), [9,12] (Ox), [10,13] (Nm)	15% (3/20) REF. [15] (Ox), [16] (Ox/Cef), [19] (Nd)	5% (1/20) REF. [17] (Nm)
DDM	10% (2/20) REF. [18,21] (Cef)	30% (6/20) REF. [24,25,26] (Cef), [22] (Ox), [23] (Ox/Cef), [20] (Nm)	0% (0/20)
Total (100%)	50%	45%	5%

## Data Availability

The data presented in this study are available in this article (and Appendix A).

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
