# Peer review of "Differences in Biofilm Formation by Methicillin-Resistant and Methicillin-Susceptible Staphylococcus aureus Strains"

_diseases, 2023, doi:10.3390/diseases11040160_

Round 1
Reviewer 1 Report
Comments and Suggestions for Authors
Point 1:Please label the references in a unified format and order.
Point 2:The latest literature is too few, with only 15 articles in the past five years (2018-2023,15/65=0.23). Please add some latest literature.
Point 3: Line 16 - "Staphylococcus aureus..." -> Staphylococcus aureus (S. aureus)...;
Point 4: Line 54 - "...mecA..." -> ...mecA...;
Point 5: Line 80 - "...28.2" -> ...28.2%;
Point 6: Line 116 - "...S. epidermidis..." -> ...Staphylococcus epidermidis (S. epidermidis)...;
Point 7: Line 151 - "...BHI NaCl..." -> ...BHI-NaCl...;
Point 8: Line 200 - "...Protein A (spa) ..." -> ...Protein A (Spa)...;
Point 9: Line 240 - "...a S. aureus MRSA clinical isolate ..." -> This sentence refers to S.aureus or MRSA
Point 10: In Figure 1 and Figure 2, please delete the “Created in BioRender.com bio”.
Point 11: Line 297 - "...sarA..." -> ...SarA...;
Point 12: Line 327,331 - "...sarA ..." -> ...SarA...;
Point 13: Gene name needs to be italicized.
Comments on the Quality of English Language
English language fine. No issues detected
Author Response
Dear Professor, thank you for revising our manuscript. We have taken into consideration all your comments and suggestions. We indicate the changes you proposed in the new version of the revised manuscript.
Point 1:Please label the references in a unified format and order.
Thank you. We follow the instructions for listing the references in agreement with the journal citation style.
Point 2:The latest literature is too few, with only 15 articles in the past five years (2018-2023,15/65=0.23). Please add some latest literature.
We added some recent literature in this revised version of the manuscript (4, 5, 27, 28, 29, 30, 32, 33, 69, 70). Some old references were important to explain the first publications describing for the first time PIA-dependent biofilm formation in S. epidermidis, the first associations of the mecA presence with the biofilm phenotype, and the first PIA-independent mechanisms in biofilm formation for S. aureus strains.
Point 3: Line 16 - "Staphylococcus aureus..." -> Staphylococcus aureus (S. aureus)...;
We have changed to the correct form and the abbreviation was included. Line 16.
Point 4: Line 54 - "...mecA..." -> ...mecA...;
We have changed to the correct form. Line 62.
Point 5: Line 80 - "...28.2" -> ...28.2%;
We have added the percentage symbol. Line 97.
Point 6: Line 116 - "...S. epidermidis..." -> ...Staphylococcus epidermidis (S. epidermidis)...;
We have included the non- and abbreviated form.Line 161.
Point 7: Line 151 - "...BHI NaCl..." -> ...BHI-NaCl...;
The correction was done.198.
Point 8: Line 200 - "...Protein A (spa) ..." -> ...Protein A (Spa)...;
The correction was done. Line 246.
Point 9: Line 240 - "...a S. aureus MRSA clinical isolate ..." -> This sentence refers to S.aureus or MRSA
It refers to MRSA. We have deleted ¨S. aureus¨ to avoid confusion. Line 287.
Point 10: In Figure 1 and Figure 2, please delete the “Created in BioRender.com bio”.
Thank you! We agree with you. We have removed that label from both figures. They look much better. Thank you!
Point 11: Line 297 - "...sarA..." -> ...SarA...;
The correction was done. Line 344.
Point 12: Line 327,331 - "...sarA ..." -> ...SarA...;
The correction was done. Line 374, 378.
Point 13: Gene name needs to be italicized.
Thank you! This was a common observation made also by the other reviewers. We have checked the gene and protein names according to the nomenclature.

Reviewer 2 Report
Comments and Suggestions for Authors
The present review aimed to discuss differences between methicillin-susceptible (MSSA) and methicillin-resistant Staphylococcus aureus (MRSA).
Some issues need to be clarified:
General comments
- Only 20 articles were selected for the review. What criteria were used to select the articles that made up the review? Where was the article search performed?
- Once the criteria have been defined, if only 20 articles remain, a supplementary table with article data such as number of isolates analyzed, origin of isolates, culture medium used, biofilm detection system, and for MRSA which type of SCCmec, will add value to the review.
- From item 1.2 onwards, the authors discussed characteristics of biofilms dependent and non-dependent on the icaA operon, the role of the agr system in biofilm regulation, but the differences between MSSA and MRSA were not clearly presented.
-The authors should present some data more completely. For example, in the lines 299-302: the number of up- and down-regulated genes was mentioned, but in general these genes participate in which main pathways related to biofilm?
Specific comments
-Table 1: The presentation of data in table 1 needs to be improved. And, as mentioned previously, more information about the selected studies should be added as Supplementary Table.
-Lines 80-82: “As expected, in the work with the lowest percentage of agreement, it was found that MRSA and S. aureus isolates containing the mecA gene were better biofilm producers than MSSA isolates only for the Congo red assay but not for the microtiter method”. Please explain “S. aureus isolates containing the mecA gene” . Does this refer to oxacillin-susceptible mecA positive S. aureus (OS-MRSA)? Moreover, a brief explanation of the principle of the Congo red test should be introduced at the first mention of the test. This information is important for laypeople to be able to analyze the differences observed between the tests.
-Figure 1: The role of each component of the icaADBC operon was not presented in the main text or in the caption.
Lines 125-126: “This may explain, at least in part, why not all the S. aureus isolates containing those genes can form biofilms.” This sentence wants to explain that the expression of the ica A operon is “is tightly regulated in vitro”? Based on the previous information (lines 111-124), the authors did not present data that contributes to this statement. This statement could be placed in lines 240-247.
Lines 164-165: “These results clearly show differences in biofilm formation by MSSA and MRSA strains depending on the composition of the medium”. Again, this statement highlighting differences in the culture medium as "responsible for the differences between MSSA and MRSA biofilms" is not appropriate.
Lines 340-341: “We have addressed in this review recent literature comparing the ability of biofilm 340 formation by MRSA and MSSA isolates”. Please evaluate this information, as most of the articles presented as References are not classified as recent.
Minor comments
Line 142: please remove “S. aureus”
Please check whether in the text you are referring to the gene or its product (for example, protein). According to the nomenclature of bacterial genes and their phenotypes: Gene – lowercase italic; Phenotypes – the first letter of the symbol is capitalized and not italicized.
Line 246: please correct “Sortase”
Line 322: please correct “respond”
Lines 337-338: “Figure 1 and Figure 2 show the regulation of the icaADBC operon and agrBDCA operon, respectively.” This sentence can be removed or moved next to the figures.
Comments on the Quality of English LanguageMinor editing of English language required.
Author Response
The present review aimed to discuss differences between methicillin-susceptible (MSSA) and methicillin-resistant Staphylococcus aureus (MRSA).
Some issues need to be clarified:
General comments
- Only 20 articles were selected for the review. What criteria were used to select the articles that made up the review? Where was the article search performed?
Dear Professor, we agree with you. We did not indicate that in the manuscript. We searched for articles analyzing biofilm formation in MRSA and MSSA in the databases PubMed, Web of Science, and Scopus. We only found 20 articles comparing with similar conditions the biofilm formation capacity by those strains. Even though the number of articles seems to be low, we found interesting the analysis and information of these relatively recent publications. Please consider that in the next section (Differences in biofilm formation by MRSA and MSSA), the comparison continues, but this time analyzing the changes in the composition of medium and the biofilm phenotype by both types of strains.
We have included the next explanation in the introduction section: ¨We have searched in the databases PubMed, Scopus, and Web of Science for publications analyzing the differences in biofilm formation by MRSA and MSSA isolates using similar conditions. ¨
- Once the criteria have been defined, if only 20 articles remain, a supplementary table with article data such as number of isolates analyzed, origin of isolates, culture medium used, biofilm detection system, and for MRSA which type of SCCmec, will add value to the review.
At first, we tried to do a more extensive table, however, because of all the parameters, it was difficult to visualize the associations. Among the parameters are: 1) MRSA vs MSSA biofilm capacity, 2) MRSA classification based on mecA detection by PCR and/or Disk diffusion method (DDM), 3) Antibiotics used in the DDM (Cefoxitin and/or Oxacillin), 4) Biofilm detection method (Crystal violet staining and/or Congo Red Method), 5) Medium used to promote the in vitro biofilm formation 6) Number and source of S. aureus isolates.
In agreement with your observation, we did a modification in table 1 to include the first three parameters, and we included a supplementary table 1 to add the rest of the parameters (4,5, and 6). In relation with the SSCmec type, we added a paragraph in the introduction. Lines 129-148.
- From item 1.2 onwards, the authors discussed characteristics of biofilms dependent and non-dependent on the icaA operon, the role of the agr system in biofilm regulation, but the differences between MSSA and MRSA were not clearly presented.
The main differences in biofilm formation by MRSA and MSSA isolates have been described in vitro. For example, under osmotic stress conditions, MSSA tend to form PIA-dependent biofilms, because of the icaADBC operon expression is favored (this is explained in section 1.1). MRSA isolates tend to form protein- and extracellular DNA-dependent biofilms favored by the addition of glucose to the medium. One critical experiment that was described to analyze the composition of biofilms by MRSA and MSSA, was the use of proteinase K or sodium metaperiodate that destabilize proteins or β-1,6-linked polysaccharides, respectively (this was explained in section 1.1).
In case of the MRSA protein-dependent biofilms. LPXTG surface proteins such as FnbA/B and protein A (Spa) have been described to be important. We also highlighted the role of some hydrolases in the bacterial cell lysis and release of eDNA (this was explained in section 1.2).
Also, the Agr system that enhanced the secretion of proteases and inhibits the expression of surface proteins has been shown to impair biofilm formation only in MRSA but not in MSSA, in agreement with a protein-dependent biofilm phenotype in MRSA isolates (this is explained in section 1.3).
-The authors should present some data more completely. For example, in the lines 299-302: the number of up- and down-regulated genes was mentioned, but in general these genes participate in which main pathways related to biofilm?
Dear Professor, as SarA induces the expression of the agr operon, the previous idea was that the role of SarA in biofilm formation was exclusively through the regulation of the Agr system. In this specific case, we only mentioned the differences in the up- and down-regulated genes by the same sarA and agr mutant strains to introduce the concept of SarA regulation of biofilm formation independent of Agr. As you can see, this idea continues with other more specific examples of genes participating in biofilm formation in the same paragraph. This idea is important as SarA participates in both biofilm phenotypes 1) PIA-dependent and 2) protein-dependent biofilms, while Agr only participates in some MRSA strains with protein-dependent biofilm phenotype (section 1.3).
Specific comments
-Table 1: The presentation of data in table 1 needs to be improved. And, as mentioned previously, more information about the selected studies should be added as Supplementary Table.
As mentioned before, we modified table 1 and included a supplementary figure containing the information that you kindly suggested.
-Lines 80-82: “As expected, in the work with the lowest percentage of agreement, it was found that MRSA and S. aureus isolates containing the mecA gene were better biofilm producers than MSSA isolates only for the Congo red assay but not for the microtiter method”. Please explain “S. aureus isolates containing the mecA gene” . Does this refer to oxacillin-susceptible mecA positive S. aureus (OS-MRSA)?
We understand the confusion. In that specific publication by Leshem T (doi: 10.1111/jam.15612), the biofilm formation was compared for MRSA and MSSA using the classification of Disk Diffusion Method (Table 2), but also by analyzing the biofilm formation in 79 clinical isolates and its association with the mecA presence by PCR (Table 5).We agree with you and to avoid confusion we shorten the sentence as follows:¨As expected, in the work with the lowest percentage of agreement, it was found that MRSA were better biofilm producers than MSSA isolates only for the Congo red assay but not for the microtiter method. ¨
Moreover, a brief explanation of the principle of the Congo red test should be introduced at the first mention of the test. This information is important for laypeople to be able to analyze the differences observed between the tests.
Thank you for the advice. We added the next paragraph to briefly explain the biofilm detection methods:
¨The Congo Red Method is a qualitative assay for detection of biofilm producer bacteria. The method uses Congo Red dye and sucrose in BHI agar. Bacteria forming black colonies with a dry and crystalline appearance are considered biofilm producers, while those with pinkish-red colonies are consider non-biofilm producers. In case of the microtiter plate assay, bacteria are allowed to form biofilms in 96-well polystyrene microplates, then the biofilm is stained with Crystal Violet dye. Finally, the biofilm can be quantified by dissolving it with acetic acid and checking the optical density at around 600 nm in a plate reader (Ref. 27). ¨
-Figure 1: The role of each component of the icaADBC operon was not presented in the main text or in the caption.
We completely agree with that. We forgot to add that information in the text. We have added the next paragraph in the main text to address that point.
¨IcaA is a transmembrane N-acetyl-glucosaminyltransferase that together with IcaD synthesize N-acetyl-glucosamine oligomers of more than 20 residues. Then, IcaC translocates these polysaccharides out of the cell surface. Finally, IcaB deacetylates the poly-N-acetylglucosamine molecules which is important for the adherence of the polymers to the outer cell surface and biofilm formation (Ref. 33). ¨
Lines 125-126: “This may explain, at least in part, why not all the S. aureus isolates containing those genes can form biofilms.” This sentence wants to explain that the expression of the ica A operon is “is tightly regulated in vitro”? Based on the previous information (lines 111-124), the authors did not present data that contributes to this statement. This statement could be placed in lines 240-247.
We agree with that. We have added that information in the introduction:
¨We now know that icaADBC genes are present in most of the S. aureus clinical isolates, and that expression of those genes is tightly regulated in vitro, for example, CO2 levels, anaerobicity, glucose, and osmotic stress have been shown to influence ica operon expression and biofilm formation (Ref. 19,37).
Lines 164-165: “These results clearly show differences in biofilm formation by MSSA and MRSA strains depending on the composition of the medium”. Again, this statement highlighting differences in the culture medium as "responsible for the differences between MSSA and MRSA biofilms" is not appropriate.
Dear Professor, we decided to eliminate that sentence. In general, MSSA tend to form PIA-dependent biofilms, and MRSA, protein-dependent biofilms.
Lines 340-341: “We have addressed in this review recent literature comparing the ability of biofilm 340 formation by MRSA and MSSA isolates”. Please evaluate this information, as most of the articles presented as References are not classified as recent.
Dear professor, we mentioned ¨recent¨ only to refer the relatively recent 20 publications in the comparison analysis in the introduction section. 2006 (1), 2007 (2), 2013 (1), 2014 (1), 2016 (1), 2017 (1), 2018 (1), 2019 (2), 2020 (3), 2021 (3), 2022 (3), and 2023 (1). However, to avoid confusion we deleted the word ¨recent¨. Please consider that some old references were critical for this review as they explained the first description of PIA-dependent biofilms found in S. epidermidis, and the first association of methicillin-resistance and mecA gene also in S. epidermidis. After this, the attention was on S. aureus and MRSA strains as they have more clinical relevance. Also, some references mentioning the first descriptions of PIA-independent biofilm formation in S. aureus.
Minor comments
Line 142: please remove “S. aureus”
We have deleted ¨S. aureus¨ for a better understanding. Line 189.
Please check whether in the text you are referring to the gene or its product (for example, protein). According to the nomenclature of bacterial genes and their phenotypes: Gene – lowercase italic; Phenotypes – the first letter of the symbol is capitalized and not italicized.
Thank you! We have checked and written the genes and proteins according to the nomenclature.
Line 246: please correct “Sortase”
We have made the correction. Line 293.
Line 322: please correct “respond”
We have changed to ¨response¨. Line 368.
Lines 337-338: “Figure 1 and Figure 2 show the regulation of the icaADBC operon and agrBDCA operon, respectively.” This sentence can be removed or moved next to the figures.
We have deleted that sentence from both figures.

Reviewer 3 Report
Comments and Suggestions for Authors
In this review article, the authors discuss biofilm formation by Staphylococcus aureus and address the question of whether there are important differences in the capacity of methicillin-resistant strains relative to those of methicillin-sensitive strains to produce biofilms, an important virulence attribute of this species. This is clearly an interesting question and the authors additionally discuss features of biofilm formation. The authors tabulated MRSA versus MSSA frequencies from recent publications and from this concluded that “Even though the results proposed that MRSA strains are better in biofilm formation than their counterpart MSSA”. However, the approach used by the authors to quantify the biofilm phenotype is fundamentally flawed and, therefore the numbers are unreliable. The flaws are as follows:
1. The biofilm phenotype is entirely based on in vitro assays. There is no evidence that a biofilm-negative strain as assessed in vitro is biofilm-negative in vivo. As evidence of this, the authors discuss reference 48, and in this study differences in biofilm formation in vitro were documented despite all of the strains being clinical isolates from device-related infections. Thus, by definition all of the strains would have been biofilm-formers in vivo.
2. Included in the Table 1 references were studies with bovine strains (for example reference 1), and large animal (LA) strains of S. aureus are distinct from human clinical strains and this confounds the comparison. Better to restrict the comparison to strictly human strains.
3. The references cited in Table 1 evaluated biofilm formation and antibiotic resistance. They usually did not further characterize they strains with regard to MLST type or spa type. As a consequence, dominant strains in circulation at that particular geographic region would likely be present multiple times in the sample population. Thus, a hot clinical strain would be over-represented and likely skew the biofilm phenotype numbers. This would be more probable with the MRSA strains which tend to be more clonal. The only reliable studies that should be used to assess biofilm frequencies would be those which more rigorously type the isolates to eliminate redundant strains in the studied groups.
Because the major point of this review is to provide evidence for difference is biofilm formation between MRSA and MSSA strains and to discuss reasons why this may be the case, the premise is unsupported and the review is misleading.
Other comments:
1. Lines 80-83: Since you bring of this point, perhaps some more detail on the two biofilm assays and the benefits and limitations of each should be given.
2. Lines 86-91: This would be an issue if many strains that are classified as resistant by the disc test are found to lack the mecA determinant. If so, this should be discussed here and whatever information is available beside reference 19.
3. Lines 100-101: In reference 19, the authors detected the mecA determinant by PCR. Therefore, they tested for the presence of this gene, but provided no documentation of its “expression”.
4. Lines 157-165: You are drawing conclusions based on studies that had two unrelated variables, namely MRSA and biofilm production in BHI-glucose and MSSA and biofilm production in BHI-NaCl. You cannot state that differences were because one was MRSA and the other MSSA.
5. Lines 189-196: What is your point here? How does the presence of these genes relate to biofilm formation?
6. Lines 276-278: You need a reference citation for this statement.
7. Lines 278-280: This is just a restatement of the previous sentence, not a second observation. The agr-negative strains are assumed to be agr mutants. If not, more detail is needed here.
8. Lines 290-292: Why is this interesting? The different systems operate in the same manner.
9. Lines 293-295: Why is this “on the other hand”? The above paragraph indicates agr activation disrupts biofilm production by protease production etc. and 6390 lacking agr is a more stable biofilm producer, which is the expected outcome.
10. Lines 295-297: This statement is confusing, especially with the “however”. SarA is an inhibitor of agr expression, so mutants lacking SarA would be expected to have higher agr activation and consequently a reduced biofilm phenotype, independently of other regulatory effects. It also has other effects, as noted in the following section of the manuscript.
11. Lines 345-346: Need references to support this claim that virulence factor differences between MRSA and MSSA strains are controversial.
12. Lines 349-354: Actually, this classification is still useful because community isolates tended to be methicillin-resistant but not broadly antibiotic resistant whereas hospital-associated isolates were multiply antibiotic resistant, in addition to virulence factor content differences.
13. Lines 357-361: I disagree strongly with this conclusion, as indicated in comment #4, above.
Comments on the Quality of English LanguageMinor issues, such as repeated use of "on the other hand".
Author Response
In this review article, the authors discuss biofilm formation by Staphylococcus aureus and address the question of whether there are important differences in the capacity of methicillin-resistant strains relative to those of methicillin-sensitive strains to produce biofilms, an important virulence attribute of this species. This is clearly an interesting question and the authors additionally discuss features of biofilm formation. The authors tabulated MRSA versus MSSA frequencies from recent publications and from this concluded that “Even though the results proposed that MRSA strains are better in biofilm formation than their counterpart MSSA”. However, the approach used by the authors to quantify the biofilm phenotype is fundamentally flawed and, therefore the numbers are unreliable. The flaws are as follows:
- Dear Professor, thanks for revising our manuscript. We appreciate your time and guidance. The changes are indicated in the new revised version of the manuscript.
- The biofilm phenotype is entirely based on in vitro assays. There is no evidence that a biofilm-negative strain as assessed in vitro is biofilm-negative in vivo. As evidence of this, the authors discuss reference 48, and in this study differences in biofilm formation in vitro were documented despite all of the strains being clinical isolates from device-related infections. Thus, by definition all of the strains would have been biofilm-formers in vivo.
- Dear Professor, we agree with you in this point. There is a limitation in the literature describing biofilm formation in vivo. I consider that it may be because of the challenging of these types of experiments in animal models. The in vitro experiments may be helpful and reliable only for some device-related infections. For example, most of the experiments are carried out in polystyrene microplates, and this surface may resemble the surface of catheters. The surface and microenvironment must be important for biofilm formation of bacteria. However, as we mentioned, there was also a limitation in the description of biofilm formation in other surfaces such as fibrinogen or collagen, extracellular matrix components that may be related to wound or soft tissue infections by aureus.
- Included in the Table 1 references were studies with bovine strains (for example reference 1), and large animal (LA) strains of S. aureus are distinct from human clinical strains and this confounds the comparison. Better to restrict the comparison to strictly human strains.
- We agree with that. There were only 2 publications in the table that are not clinical isolates. One contains aureus isolates form cattle and the other from food commodities. We have removed them from the analysis. We included reference 19 and 20 to complement the table. We haven’t included these 2 publications before because they analyzed biofilm formation by MRSA and MSSA using different media composition. We only include data using BHI-glucose as most of the publications in the table use this condition.
- Unfortunately, we did not find many publications analyzing the differences in biofilm formation by both strains. We searched in PubMed, Web of Science, and Scopus. This searching criterion is now indicated in the introduction of this revised manuscript version. Based on your comments, we also mentioned in the introduction that this is a limitation in the analysis.
- The references cited in Table 1 evaluated biofilm formation and antibiotic resistance. They usually did not further characterize they strains with regard to MLST type or spa type. As a consequence, dominant strains in circulation at that particular geographic region would likely be present multiple times in the sample population. Thus, a hot clinical strain would be over-represented and likely skew the biofilm phenotype numbers. This would be more probable with the MRSA strains which tend to be more clonal. The only reliable studies that should be used to assess biofilm frequencies would be those which more rigorously type the isolates to eliminate redundant strains in the studied groups.
Because the major point of this review is to provide evidence for difference is biofilm formation between MRSA and MSSA strains and to discuss reasons why this may be the case, the premise is unsupported and the review is misleading.
- Dear Professor, we understand your point emphasizing that some clones would have been dominant in a specific place and time. Most of the articles that we analyzed for this review did not carry out the MLTS typing that you mentioned. We though it was a good idea to share this information that you mentioned in order that, in the future, these types od analysis of virulence factor or biofilm capacity must take into consideration the origin of the aureus isolates.
- Line 29-148. ¨Even though there seems to be a tendency of MRSA to be better in biofilm formation in comparison with MSSA isolates, only when the presence of mecA was determined to classify the aureus isolates as MRSA, the few research articles analyzing these differences is a limitation in the analysis. Also, the results may be validated with a more precise lineage classification of the isolates, as a dominant clone may be present during the time of the study in a specific region. For example, there have been described for S. aureus several clonal complexes with and endemic or global distribution based on the differences in the sequence of seven housekeeping genes (arcC, aroE, glpF, gmk, pta, tpi, and yqiL) by multi-locus sequence typing (MLST) (S. aureus clonal complex designation | PubMLST) [30]. These clonal complexes, depending on the type, may include community-associated MRSA (CA-MRSA), hospital-associated MRSA, or MSSA clones. In addition, the mecA gene resides in a genetic mobile element called staphylococcal cassette chromosome (SCCmec), and to date, there have been recognized 14 types of SCCmec elements by the International Working Group on the Classification of Staphylococcal Cassette Chromosome Elements based on genetic variation of the two main components, the ccr gene complex and the mec gene [31]. Some SCCmec types have been associated with HA-MRSA (SCCmec type I, II, and III)., while others with CA-MRSA (SCCmec type IV and V) [32]. Therefore, the association analysis between MRSA and MSSA isolates in relation to the biofilm formation capacity, must include in the future, a better characterization of the isolates in order to provide more information related to their clonal origin. ¨
Other comments:
- Lines 80-83: Since you bring of this point, perhaps some more detail on the two biofilm assays and the benefits and limitations of each should be given.
We added the next paragraph in the introduction. Thank you!
- Lines 80-87. ¨The Congo Red Method is a qualitative assay for detection of biofilm producer bacteria. The method uses Congo Red dye and sucrose in BHI agar. Bacteria forming black colonies with a dry and crystalline appearance are considered biofilm producers, while those with pinkish-red colonies are consider non-biofilm producers. In case of the microtiter plate assay, bacteria are allowed to form biofilms in 96-well polystyrene microplates, then the biofilm is stained with Crystal Violet dye. Finally, the biofilm can be quantified by dissolving it with acetic acid and checking the optical density at around 600 nm in a plate reader ( 27). ¨
- Lines 86-91: This would be an issue if many strains that are classified as resistant by the disc test are found to lack the mecA determinant. If so, this should be discussed here and whatever information is available beside reference 19.
- There are some methicillin-resistant aureus strains whose resistance to this antibiotic is independent of the mecA gene. As it has been suggested an association of the mecA presence with the biofilm phenotype, we consider that it may be relevant to analyze this. We have included an explanation of this point in the text.
- Lines 117-121. ¨It is generally accepted that methicillin resistance is acquired by the presence of mecA; however, there are descriptions of methicillin resistance independent of mecA such as borderline oxacillin-resistant aureus (BORSA) strains and modified methicillin-resistant S. aureus (MODSA) strains, but the frequency and clinical relevance is uncertain [28,29]. ¨
- Lines 100-101: In reference 19, the authors detected the mecA determinant by PCR. Therefore, they tested for the presence of this gene, but provided no documentation of its “expression”.
- Yes, it was only analyzed the presence of the mecA It was not analyzed its expression. We have changed ¨expression¨ for ¨presence¨ in that sentence to avoid confusion. Line 116.
- Lines 157-165: You are drawing conclusions based on studies that had two unrelated variables, namely MRSA and biofilm production in BHI-glucose and MSSA and biofilm production in BHI-NaCl. You cannot state that differences were because one was MRSA and the other MSSA.
- Dear Professor, we understand your point. We removed the next sentence ¨These results clearly show differences in biofilm formation by MSSA and MRSA strains depending on the composition of the medium. ¨ Also, we modified the conclusion based on your comment. In general, MSSA isolates have been associated with a PIA-dependent biofilm phenotype and MRSA isolates with a protein-dependent biofilm phenotype. Also, in the next sections, is is described the genes involved in the protein-biofilm phenotype.
- Lines 189-196: What is your point here? How does the presence of these genes relate to biofilm formation?
- It was first described the PIA-dependent biofilm phenotype (β-1,6-linked polysaccharide) in epidermidis and then in S. aureus. However, later become clear that there must be proteins involved in the PIA-independent biofilm phenotype. The point of mentioning the genes found in the genetic analysis was introductory of the proteins involved in the proteinaceous- and eDNA-dependent biofilm phenotype associated mainly with MRSA. For example, LPXTG proteins and hydrolases, involved in cell lysis and release of eDNA. The next paragraph in the manuscript describes these genes in more detail.
- Lines 276-278: You need a reference citation for this statement.
- Thank you! It was included. Ref. 59.
- Lines 278-280: This is just a restatement of the previous sentence, not a second observation. The agr-negative strains are assumed to be agr mutants. If not, more detail is needed here.
- We modified a little the sentence for a better understanding. ¨Analyzing the agr phenotype in terms of δ-toxin production (the product of the RNAIII transcript in the agr operon) and the biofilm formation capacity, it was demonstrated that 78% of the agr-negative aureus strains formed biofilm in comparison with only 6% of the agr-positive strains¨
- In the next sentence the mutant refers to agr-deletion in the strains analyzed.
- Lines 290-292: Why is this interesting? The different systems operate in the same manner.
- There is a typing based in genetic variations of the agr genes (Agr type I, II, III and IV). There has been an association of some types with specific geographical locations of the aureus isolates. However, we agree with you, and we have removed this sentence to simplify the explanation.
- Lines 293-295: Why is this “on the other hand”? The above paragraph indicates agr activation disrupts biofilm production by protease production etc. and 6390 lacking agr is a more stable biofilm producer, which is the expected outcome.
- We agree with you. It is not a contrasting sentence in relation with the previous paragraph. We have changed to ¨Furthermore¨. We apologize for that!
- Lines 295-297: This statement is confusing, especially with the “however”. SarA is an inhibitor of agr expression, so mutants lacking SarA would be expected to have higher agr activation and consequently a reduced biofilm phenotype, independently of other regulatory effects. It also has other effects, as noted in the following section of the manuscript.
- SarA positively regulates Agr system. This system activates the expression of proteases and inhibits surface proteins, thus, disrupting protein-dependent biofilms. For the strain mentioned, that is the reason of the biofilm formation with the deletion of the agr But when a second sarA-mutation was introduced in this agr-mutant, the biofilm-phenotype was lost indicating a relationship between SarA and the Agr system.
- The most important of the sentence is to start making a connection of SarA and the Agr system. Later, in the paragraph we explain the role of SarA in biofilm formation independent of this system. This is important as SarA participates in both biofilm phenotypes: PIA-dependent and protein-dependent biofilms, while the Agr system negatively modulate the protein-dependent biofilms. Please have a look of the graphical abstract in this new version. Even though SarA activates the expression of the agr operon, they have also independent and contrasting effects in modulating the activity of proteases and surface proteins. –
- In this case we consider that the sentence is well explained.
- Lines 345-346: Need references to support this claim that virulence factor differences between MRSA and MSSA strains are controversial.
- We have added some references to address this point in the introduction where it was also mentioned. Lines 68-71. Ref. 5 and 6.
- Lines 349-354: Actually, this classification is still useful because community isolates tended to be methicillin-resistant but not broadly antibiotic resistant whereas hospital-associated isolates were multiply antibiotic resistant, in addition to virulence factor content differences.
- Thanks for this explanation. We explained this in the introduction together with the identification of the clonal complexes by MLTS, as you suggested. We also modified the conclusions based on your comment.
- Lines 357-361: I disagree strongly with this conclusion, as indicated in comment #4, above.
- We modified the conclusion to explain only the tendency of MSSA to form PIA-dependent biofilms and MRSA protein-dependent biofilms.

Reviewer 4 Report
Comments and Suggestions for Authors
In this manuscript, authors perform an extensive literature review in order to identify interactions and relations regarding methicillin-resistance, biofilm-formation, biofilm nature, implication of agr-system and other regulator systems in S. aureus. In my opinion, the review is well organized and presented in detail.
Remarks
I think it is more precise to use the term susceptible instead of sensitive for MSSA, so it would be methicillin-susceptible
references: are they in compliance to journal's guidelines?
line 16/abstract Staphylococcus aureus italics
line 54 mecA should be written in italics
line 82 % is missing after 28.2
line 115 not italics these are proteins
line 143 Ica should be ib italics
line 293 "described" instead of "showed"
Author Response
Comments and Suggestions for Authors
In this manuscript, authors perform an extensive literature review in order to identify interactions and relations regarding methicillin-resistance, biofilm-formation, biofilm nature, implication of agr-system and other regulator systems in S. aureus. In my opinion, the review is well organized and presented in detail.
- Dear Professor. Thanks for your time revising our manuscript. Your comments and suggestions were taken into consideration. The changes are indicated based on the new version of this revised manuscript.
Remarks
I think it is more precise to use the term susceptible instead of sensitive for MSSA, so it would be methicillin-susceptible
- Thank you! We agree with that. Even though there are some articles using the word ¨sensitive¨, it is more common the word ¨susceptible¨ to describe antibiotic susceptibility. We have changed ¨sensitive¨ to ¨susceptible¨. You can see the changes in the title (line 3), abstract (line 20) and introduction (line 56).
references: are they in compliance to journal's guidelines?
- We have modified to the right format.
line 16/abstract Staphylococcus aureus italics
- We have changed to italics Line 16.
line 54 mecA should be written in italics
- We have changed to italics. Line 62.
line 82 % is missing after 28.2.
- We have added the percentage symbol. Line 97.
line 115 not italics these are proteins
- We have made that correction. Line 155.
line 143 Ica should be ib italics
- We have made that correction. Line189.
line 293 "described" instead of "showed"
- We have made that correction. Line 340.

Round 2
Reviewer 2 Report
Comments and Suggestions for Authors
Dear authors,
Thank you for all replies. There are small corrections that can be made at the proofing stage.
Line 56: please correct “SCCmec” and add staphylococcal cassette chromosome, because the definition of the acronym was only presented in line 134).
“of a foreign mobile genetic element containing mecA (staphylococcal cassette chromosome mec - SCCmec) into the bacterial chromosome”
Lines 133-138: As the definition of acronym of SCCmec was transferred to line 56, the information can be removed from these lines.
“In addition, there have been recognized 14 types of SCCmec elements by the International Working Group on the Classification of Staphylococcal Cassette Chromosome Elements based on genetic variation of the two main components, the ccr gene complex and the mec gene [31]”.
My suggestion:
“In addition, 14 types of SCCmec elements are currently recognized by the International Working Group on the Classification of Staphylococcal Cassette Chromosomal Elements based on the genetic variation of the two main components, the ccr gene complex and the mec gene [31].”
Line 133: please add “(HA-MRSA)” after hospital-associated MRSA
Line 139: please remove the point after “III),”
Line 224: please remove the capital letter from Bap (gene)
Line 248: please add italic to “fnbA” and “fnbB”
Line 287: “reveled” or “revealed”?
Line 387: please change “virulent” to “virulence”
Author Response
Dear authors,
Thank you for all replies. There are small corrections that can be made at the proofing stage.
- Dear Professor, thanks to you for taking the time to revise our manuscript. We really appreciate it; this will help to improve the final version.
Line 56: please correct “SCCmec” and add staphylococcal cassette chromosome, because the definition of the acronym was only presented in line 134).
“of a foreign mobile genetic element containing mecA (staphylococcal cassette chromosome mec - SCCmec) into the bacterial chromosome”
- We agree with you! We have now included the definition of the acronym. Line 54. ¨of a foreign mobile genetic element containing mecA called staphylococcal cassette chromosome (SCCmec), into the bacterial chromosome of MRSA strains [3]. ¨
Lines 133-138: As the definition of acronym of SCCmec was transferred to line 56, the information can be removed from these lines.
“In addition, there have been recognized 14 types of SCCmec elements by the International Working Group on the Classification of Staphylococcal Cassette Chromosome Elements based on genetic variation of the two main components, the ccr gene complex and the mec gene [31]”.
My suggestion:
“In addition, 14 types of SCCmec elements are currently recognized by the International Working Group on the Classification of Staphylococcal Cassette Chromosomal Elements based on the genetic variation of the two main components, the ccr gene complex and the mec gene [31].”
- We have modified the sentence as you have suggested. Thank you! Line 130-134.
Line 133: please add “(HA-MRSA)” after hospital-associated MRSA
- We have included the abbreviation. Line 130. ¨These clonal complexes, depending on the type, may include community-associated MRSA (CA-MRSA), hospital-associated MRSA (HA-MRSA), or MSSA clones.¨
Line 139: please remove the point after “III),”
- We have removed it. Thank you! Line 135.
Line 224: please remove the capital letter from Bap (gene)
- It was changed to the correspondent italic lowercase letter. Line 220.
Line 248: please add italic to “fnbA” and “fnbB”
- Thanks Professor! Last time, you explained very well the nomenclature rules. We apologize for those mistakes. Line 244.
Line 287: “reveled” or “revealed”?
- It is ¨revealed¨. We have changed it. Thank you! Line 283.
Line 387: please change “virulent” to “virulence”
- Thank you! We have changed to ¨virulence¨. Line 380.

Reviewer 3 Report
Comments and Suggestions for Authors
The authors have responded well to the concerns and comments raised in the prior review and the manuscript is improved as a result. However, they still conclude that MRSA strains are better biofilm producers than MSSA strains, despite the flaws inherent in the analysis. They have added comments (lines 149-142) that more detailed analysis of the strains is needed. However, to readers lacking familiarity with the clonal nature of circulating clinical strains, the take-home message from this article is that MRSA strains are better biofilm producers. This has not even remotely been shown to be the case, and so therefore, even with the added text, this remains misleading. What can be said with certainty is that biofilm formation has been documented with both MRSA and MSSA clinical isolates.
The review would be more accurate, and equally valuable, if Table 1, the text referring to it, and quantitative statements concluding MRSA is better at biofilm formation were removed. The discussion about differences in biofilm formation can stand alone as a positive contribution to the literature.
Author Response
The authors have responded well to the concerns and comments raised in the prior review and the manuscript is improved as a result. However, they still conclude that MRSA strains are better biofilm producers than MSSA strains, despite the flaws inherent in the analysis. They have added comments (lines 149-142) that more detailed analysis of the strains is needed. However, to readers lacking familiarity with the clonal nature of circulating clinical strains, the take-home message from this article is that MRSA strains are better biofilm producers. This has not even remotely been shown to be the case, and so therefore, even with the added text, this remains misleading. What can be said with certainty is that biofilm formation has been documented with both MRSA and MSSA clinical isolates.
The review would be more accurate, and equally valuable, if Table 1, the text referring to it, and quantitative statements concluding MRSA is better at biofilm formation were removed. The discussion about differences in biofilm formation can stand alone as a positive contribution to the literature.
- Dear Professor, thanks again for your time providing us feedback in this 2nd
- We understand your point related to the analysis of the 20 publications assessing the biofilm formation capacity by MRSA and MSSA. From the last revision, we forgot to modify the statement in the ¨concluding remarks¨ section. We have already omitted that information mentioning that MRSA was better in biofilm formation than MSSA. In the text, we have changed the statement to:
- ¨Based on this analysis, there is not enough evidence to support any difference in biofilm formation between MRSA and MSSA isolates. ¨ After this, we continue the explanation of the need to analyze the origin of the isolates by typing in clonal complexes and SCCmec Lines 122-138.
- We consider that this information is pertinent as there has not been studied in a review an analysis of the publications comparing the biofilm formation by MRSA and MSSA using similar conditions; even though the number of articles that we found was low. Also, your idea of analyzing the clonal origin of the isolates must be reached by the readers carrying out similar research in the future.
- We put much effort in the analysis related to the table 1, and we were advised by a reviewer to add a supplemental table. Among the parameters analyzed in both tables are: 1) MRSA vs MSSA biofilm capacity, 2) MRSA classification based on mecA detection by PCR and/or Disk diffusion method (DDM), 3) Antibiotics used in the DDM (Cefoxitin and/or Oxacillin), 4) Biofilm detection method (Crystal violet staining and/or Congo Red Method), 5) Medium used to promote the in vitro biofilm formation 6) Number and source of aureus isolates.
- We hope your comprehension as we are trying to follow the advice of the other reviewers. We hope to keep this information in the manuscript.
- We really appreciate your comments and suggestions!

Round 3
Reviewer 3 Report
Comments and Suggestions for Authors
The authors' responses have now alleviated my concerns.